# Surgical Management of Synucleinopathies

**DOI:** 10.3390/biomedicines10102657

**Published:** 2022-10-21

**Authors:** Sai Sriram, Kevin Root, Kevin Chacko, Aashay Patel, Brandon Lucke-Wold

**Affiliations:** Department of Neurosurgery, University of Florida, Gainesville, FL 32608, USA

**Keywords:** synucleinopathies, Parkinson’s disease, multiple system atrophy, dementia with Lewy bodies, deep brain stimulation, focused ultrasound, gene therapy, surgical techniques

## Abstract

Synucleinopathies represent a diverse set of pathologies with significant morbidity and mortality. In this review, we highlight the surgical management of three synucleinopathies: Parkinson’s disease (PD), dementia with Lewy bodies (DLB), and multiple system atrophy (MSA). After examining underlying molecular mechanisms and the medical management of these diseases, we explore the role of deep brain stimulation (DBS) in the treatment of synuclein pathophysiology. Further, we examine the utility of focused ultrasound (FUS) in the treatment of synucleinopathies such as PD, including its role in blood–brain barrier (BBB) opening for the delivery of novel drug therapeutics and gene therapy vectors. We also discuss other recent advances in the surgical management of MSA and DLB. Together, we give a diverse overview of current techniques in the neurosurgical management of these pathologies.

## 1. Introduction

The three members of the α-synucleinopathy family—Parkinson’s disease (PD), multiple system atrophy (MSA), and dementia with Lewy bodies (DLB)—are some of the most common and costly neurodegenerative diseases affecting neurosurgical patients [1]. Parkinson’s disease alone affects more than one million individuals in the United States [1,2,3]. In addition to the untold emotional burden, these afflictions place an economic strain upon the nation currently exceeding USD 50 billion per annum [4]. Considering Parkinson’s disease alone is projected to double in incidence rate by 2030 due to an aging population, it is essential to establish optimal management protocols for α-synucleinopathies [5]. In this regard, we aim to highlight current medical standards of practice as well as the efficacy of available neurosurgical interventions. Specifically, we will examine the role of deep brain stimulation (DBS), focused ultrasound (FUS), and recent advancements in the treatment of synucleinopathies.

### 1.1. Parkinson’s Disease

With a current incidence rate of 1–2 per 1000 individuals, Parkinson’s disease (PD) is the second most common neurodegenerative disease following Alzheimer’s disease [5,6]. The prevalence of PD significantly rises with age, increasing more than 5-fold from the sixth to ninth decades of life [7]. Furthermore, this prevalence is projected to rise drastically by 2030 due to an aging population, underscoring the necessity of understanding and advancing its management. Decades of investigation have given way to a current consensus that attributes PD to a combination of environmental and genetic factors [8]. The pathogenesis of PD is multifactorial, implicating oxidative stress, altered protein handling, and environmental mitochondrial toxins [8,9]. Of note to this review, mutations in the SNCA gene and mitochondrial dysfunction have been shown to result in α-synuclein accumulation [10,11]. In select brain regions in animal models, accumulations of α-synuclein have been identified to inhibit complex I of the mitochondria [12,13,14]. Additionally, α-synuclein toxicity is well supported to play a significant role in the pathogenesis of PD, although more evidence regarding the underlying molecular mechanism is needed [15,16]. The clinical progression of PD is also highly variable among patients [17]. However, the classic presentation is well documented to include tremors at rest, rigidity, bradykinesia, loss of postural reflexes, and shuffling gait [18]. Pharmacological management often focuses on symptomatic treatment, specifically targeting motor disturbances. Levodopa serves as a standard first-line treatment, functioning to supplement decreased dopamine [19]. Other dopamine modifying medications include NMDA receptor antagonists, muscarinic receptors, or dopamine receptor agonists. Degradation of dopamine is also targeted through catechol-O-methyltransferase inhibitors, and monoamine oxidase type B inhibitors [20]. Recently, pimavanserin, an atypical antipsychotic, has been described to reduce psychosis in PD, allowing for the treatment of some of the psychiatric comorbidities in PD1. Additional clinical symptoms of PD include disturbances in autonomic function, sleep disruptions, neuropsychiatric as well as sensory symptoms, and dementia, all of which are also managed pharmacologically with variable success [21,22,23]. Notably, dopaminergic medications, particularly at the higher doses required by patients with progressive disease, are associated with debilitating adverse effects. Unpredictable fluctuations of on and off states as well as levodopa induced dyskinesias represent significant barriers to effective treatment [24].

### 1.2. Multiple System Atrophy and Dementia with Lewy Bodies

In the other two diseases of the α-synucleinopathy family, Multiple system atrophy (MSA) and dementia with Lewy bodies (DLB), age is also the primary risk factor. In MSA, the annual incidence rate is 3 cases per 100,000 for a population older than 50 years [25]. Similarly, the incidence of DLB peaks in the sixth decade of life and has an incidence rate of 0.87 cases/1000 person-years in the general population [26]. Risk factors for MSA extend to environmental factors—including exposure to organic solvents, pesticides, metals, and monomers—and genetic factors—involving impaired variants of the enzyme encoded by COQ2 [27,28]. General risk factors for DLB include depression, anxiety, low caffeine intake, and stroke, as well as a genetic predisposition with specific APOE ξ4 alleles [29]. MSA is a unique member of the α-synucleinopathy family, as α-synuclein deposits in oligodendrocytes in lieu of neurons [30]. Consequently, the disease pathology stems from oligodendrogliopathy with myelin disruption from α-synuclein positive glial cytoplasmic inclusions (GCI), which leads to axon degeneration and eventual neuron degeneration [28]. Though the exact mechanism of their action is yet unknown, α-synuclein GCIs are required for the diagnosis of MSA and their density correlates with disease severity [31,32]. Similarly, though the precise pathogenesis of DLB is unknown, it is theorized to involve several metabolic pathways that lead to dysfunctions in mitochondria, purine metabolism, protein synthesis, energy metabolism [33], and α-synuclein deposits in neuronal Lewy bodies [34]. In MSA, nonmotor disruptions (i.e., respiratory, autonomic, and urogenital symptoms) often manifest first; however, it is commonly diagnosed only after motor dysfunction occurs [35]. These motor disruptions are sporadic in frequency and parkinsonian in nature, including many of the hallmark symptoms such as rigidity, bradykinesia, and postural instability. Furthermore, early autonomic dysfunctions are classic presentation features of MSA [36]. While the same parkinsonism extends to DLB, visual hallucinations, variable mental status, and dementia are also common features [37]. Treatment options for MSA are limited. MSA is characterized by a poor response to dopaminergic therapy, with only a transient improvement noted in 40% of patients [38]. Alternatively, the disease is often managed by nonpharmacologic strategies such as a decreased salt intake for orthostatic hypotension [36]. Conversely, DLB is effectively managed by acetylcholinesterase inhibitors as a first line treatment and responds well to classic dopaminergic therapy [39].

## 2. Deep Brain Stimulation (DBS)

### 2.1. DBS and α-Synuclein in PD

Despite promising pharmacological treatment options, α-synucleinopathies can prove to be extremely difficult to manage medically. For one, prolonged levodopa treatment is associated with significant side effects in PD such as dyskinesias and motor fluctuations between the on and off state [40]. Unfortunately, due to the progressive nature of the disease, dose escalation is an eventuality in most patients [41]. Deep brain stimulation (DBS) offers a promising surgical option where a high frequency electrode is implanted into a target structure in the basal ganglia or thalamus such as the globus pallidus internus (GPi), subthalamic nucleus (STN) or ventral intermediate nucleus (VIM) [42]. High frequency electrical stimulation is then delivered from the DBS electrode to normalize the pathologically exaggerated basal ganglia bursting activity seen in PD [43]. There is a potential for significant stimulation induced side effects such as paresthesia, involuntary motor contractions, speech impairment, and mood changes. To minimize these side effects, parameters of stimulation such as its amplitude and directionality relative to the DBS electrode are controlled in programming sessions in the weeks following initial implantation. DBS therapy often leads to significant reduction in medication regimens and significant prolongation of “on” medication periods. In one meta-analysis of 22 studies, L-dopa regimens were reduced over 50%, with dyskinesias reduced around 70%, and 70% reduction in “off” periods [44].

Despite the ability of DBS therapy to significantly improve quality of life, it is widely believed that it cannot stop or reverse the effects of α-synuclein-mediated neurodegeneration in PD. In fact, disease progression with loss of dopaminergic neurons is thought to occur rapidly, within 4 years [45]. Nevertheless, several recent works have expanded knowledge concerning the effect of DBS on disease progression, though there remains considerable debate as to whether long term DBS stimulation confers a neuroprotective effect. In one study, rats were induced to overexpress α-synuclein using intranigral injections of an adeno-associated viral (AAV) vector and were subsequently implanted with STN DBS [46]. Limb use by the animals was observed at baseline as well as after electrode implantation and after electrode stimulation for a period of 26 days and was compared to mice that were implanted but did not receive stimulation. Of note, though there was impaired contralateral limb use following the α-syn vector injection in all rats, there was no difference in limb use between stimulation and nonstimulation groups. Neurodegenerative changes were assessed using tyrosine hydroxylase staining (a marker of dopaminergic neurons), which also did not differ between stimulation and nonstimulation groups. This result suggests that DBS stimulation does not protect against the impairment nor the neurodegenerative changes that accompany α-synuclein accumulation [46]. Despite this, another very similar study using an AAV induced α-synuclein overexpression rat model implanted with STN DBS reported different findings. In particular, results of this study suggest that motor performance after 3 weeks of stimulation was significantly improved compared to rats in the nonstimulation group, even with stimulation turned off during motor testing. Tyrosine hydroxylase was also significantly increased in the stimulation group [47].

In addition, beta oscillations are a significant pathologic alteration observed in PD and are linked to some of its symptoms [48]. Recent evidence suggests that STN DBS suppresses these pathological beta oscillations in an AAV induced α-synuclein overexpression rat model, even 2 weeks after stimulation [49]. Other studies have also demonstrated the possible neuroprotective effects of DBS. Specifically, brain derived neurotrophic factor (BDNF) has come under interest recently for its roles in plasticity, neurogenesis, and neuroprotection [50]. In one study of rats injected with preformed fibrils of α-synuclein, though STN DBS did not impact α-synuclein deposition or total BDNF, rats in the stimulation group displayed restoration of striatal BDNF when compared to those in the nonstimulation group [51]. Furthermore, DBS may exert a neuroprotective effect through the inhibition of neuroinflammatory cytokines and pathways [52] and the modulation of synaptic plasticity [53]. Taken together, the findings related to α-synuclein deposition in PD reiterate both sides of thought related to DBS-induced neuroprotection (Figure 1).

One central regulator of α-synuclein accumulation is autophagy. Rapamycin, for instance, stimulates autophagy and improves clearance of α-synuclein [54]. Notably, derailments in α-synuclein and autophagy in the perioperative environment can lead to pathology. In turn, an important consideration in the surgical treatment of synucleinopathies is the effect of anesthetic on α-synuclein, which was explored in a recent preclinical study. In this study, rats were anesthetized with propofol for 4 h and were assessed for neurobehavioral and cognitive deficits and hippocampal α-synuclein deposition. Along with elevated hippocampal α-synuclein deposition and impaired autophagy, 4 h of propofol induced worse performance on the Morris water maze test and shorter freezing times in the freezing conditioning test [55]. These results have important implications in the surgical management of PD with DBS, where electrode implantation being conducted awake or under general anesthetic is highly center dependent.

### 2.2. DBS in DLB and MSA

Despite the abundance of work related to DBS and PD, there is a relative paucity of evidence regarding the indications and benefits related to DBS use in the remaining synucleinopathies. The risk of cognitive decline is a central issue in DBS, making preoperative neuropsychiatric evaluation of paramount importance. In one study of 60 patients treated with STN DBS, executive functioning was significantly reduced when compared to a control group receiving standard medical therapy but no DBS [56]. Risk factors for cognitive decline following DBS included age and larger preoperative medication doses. Given that patients with DLB and MSA are already at significant risk for dysfunctional cognition, this possible adverse effect should be carefully considered in the context of each patient being evaluated for DBS therapy. Nevertheless, there has been some investigation into its use in DLB and MSA. In one randomized, double blind crossover study of 6 DLB patients implanted with bilateral nucleus basalis of Meynert (NBM) electrodes, there was no difference between stimulation and control conditions in a variety of cognitive tasks. Importantly, the procedure was well-tolerated and there was a positive impact on neuropsychiatric inventory (NPI) scores in that study [57]. However, in another more recent phase 1 study of six patients with bilateral NBM DBS, some cognitive decline occurred following electrode implantation [58]. MSA is also thought to be largely unresponsive to DBS and may be an underlying explanation for why neuromodulation fails in some patients with PD [59,60]. In one recent systematic review of 12 studies representing 22 patients with MSA, the majority were treated with bilateral STN electrodes (*n* = 18) or bilateral GPi (*n* = 3) [61]. At a median follow-up of 12 months, though subjective improvements in bradykinesia, gait, and rigidity were accompanied by a more than 12-point reduction in UPDRS-III score, a significant 23% of patients displayed neurobehavioral or neurocognitive side effects following DBS implantation. In MSA patients, who are particularly at risk for cognitive impairments [62], this represents a significant barrier to the safety and tolerability of DBS in MSA61. Furthermore, this systematic review also included 12 patients with DLB treated with bilateral NBM DBS. Again, no significant improvements in quality of life nor cognitive measurements were appreciated in this study [61]. Taken together, this suggests that the surgical indications for DBS therapy in MSA and DLB are still poorly understood and require further investigation.

## 3. Focused Ultrasound

Focused ultrasound (FUS) therapy is an actively studied alternative to current standard treatments involving open surgeries [63]. The primary benefit of FUS therapy is its ability to induce biological effects on deeper target tissue without damaging surrounding tissues [63]. FUS therapy is achieved using a piezoelectric ultrasound transducer to deliver a FUS beam and is guided using imaging modalities such as a traditional ultrasound or Magnetic Resonance Imaging (MRI) [64] to provide simultaneous monitoring of tissue effects [63]. The FUS beam is steered with precision by mechanically manipulating the transducer [64], and the spatial specificity of the beam and depth of its effects can be parametrized by varying the delivered sonication frequency and intensity [65]. At high intensity, the delivered FUS beam induces two main effects on target tissue: thermoablation and cavitation. Thermoablation results from tissue absorption of the beam energy, which rapidly increases the tissue temperature to irreversibly cytotoxic levels [66]. FUS mechanically induces cavitation, or the creation of gas cavities, in tissue by expanding and compressing the tissue as it travels through it [66]. These effects are purposely leveraged in clinical treatments to target varying tissues of interest as an alternative approach to surgery. Previous research demonstrates the use of FUS thermoablation to treat uterine fibroids [67], advanced stage renal malignancy [68], and primary bone tumors [69], and thus may offer a potential alternative to contemporary invasive surgeries. FUS has emerged as a new modality for treating movement disorders—such as essential tremor and PD—through noninvasive lesioning. Additionally, FUS therapy does not require hardware placement, such as an electrode, minimizing the risk of perioperative infection. However, FUS therapy faces limitations such as attenuation from overlying tissue, sensitivity to patient movement, possible treatment times of up to several hours, and the fact that lesioning is permanent and irreversible [64,70].

### 3.1. FUS and Synuclein

One emerging field of research utilizing FUS therapy involves leveraging cavitation to open the blood–brain barrier (BBB) [71]. The BBB historically has been a major limiting factor in drug delivery to brain parenchyma [71,72] as it is only permeable to lipid-soluble molecules smaller than 400 Da [72], which restricts pharmaceutical therapy options for neurological disorders. However, recent research has shown that FUS cavitation, in conjunction with localized microbubble injections, has the ability to noninvasively and reversibly open the BBB at specific targets in vivo, providing the possibility for localized drug therapies [71,73]. In a recent phase 1 trial in PD patients, the use of FUS in the parieto-occipito-temporal junction resulted in no serious adverse effects post-treatment but exhibited an opening of the BBB, demonstrating the feasibility of this approach to enhance drug delivery [73]. In another study, FUS cavitation was used to open the BBB in mice substantia nigra and striatal caudoputamen and was coupled with intravenous administration of the potentially neuroprotective and pro-dopaminergic neurturin neurotrophic factor (NTN). It was demonstrated that, in both targeted locations, NTN was successfully delivered into brain parenchyma with minimal diffusion to nontargeted areas [71]. Though NTN bioavailability was assessed, cognitive outcomes were not reported [71]. These drug delivery results mirror the effects observed in another study that utilized localized FUS to open BBB to deliver anti-α-syn antibodies in the left hippocampus, caudate putamen, and substantia nigra of PD-model mice [74]. Importantly, the α-synuclein load was decreased without impairing neuronal cell count [74]. Another recent study utilized FU to enhance delivery of copper nanoparticles (Cu-NPs) targeted to open TRPV1 channels [75]. The opening of TRPV1 channels is proposed to induce a Ca^2+^-dependent signaling cascade, culminating in improved phagocytosis and elimination of a-syn. In this study, FU-mediated delivery of Cu-NPs ameliorated the histopathological alterations in tyrosine hydroxylase, glial fibrillary acidic protein, and α-syn. Importantly, motor, memory, and anxiety tests in mice initially worsened by α-syn aggregation were also improved with Cu-NPs [75]. Collectively, these studies demonstrate the ability of FUS cavitation to safely and effectively disrupt the BBB in vivo to facilitate targeted drug delivery into the brain parenchyma. Thus, with future research, FUS therapy coupled with localized drug delivery is an optimistic noninvasive therapeutic option that can be utilized in treating neurodegenerative diseases such as PD (Figure 2).

Beyond opening the BBB and facilitating drug delivery, FUS therapy offers potential symptomatic management options for treating PD with thermoablation. To date, there are three primary approaches of FUS therapy to treat PD symptoms: thalamotomy, subthalamotomy, and pallidotomy [76]. In one study unilaterally targeting the ventral intermediate nucleus (VIM) of the thalamus in PD patients with medication-resistant tremor, tremor was completely abolished immediately following the treatment [77]. The procedure was met with mild transient adverse effects such as headache [77]. Another recent study investigated the ability of FU therapy to abate parkinsonian symptoms by unilaterally targeting the subthalamic nucleus in PD patients with asymmetric parkinsonism [78]. Here, in a cohort of 40 patients treated with FUS or sham procedure of the STN, FUS was delivered until adequate control of tremor symptoms was achieved. UPDRS III score improved by almost 10 points at 4 months in the treatment group, which was significantly different from the 1.7-point improvement in the sham group. Specifically, FUS performed well in improving rigidity and tremor. Notably, dyskinesia, weakness, and abnormalities in gait and speech were common adverse effects. In line with previous reports, there were transient side effects related to treatment such as headache which resolved within a brief period following treatment. Importantly, there were no major cognitive or behavioral complications that developed as a result of thermoablation when measured at 4 months post-treatment [78]. FUS therapy has also been used for unilateral pallidotomy of the globus pallidus internus to improve symptoms in dyskinesia-dominant PD, again associated with transient side effects such as minor headache [79]. Overall, these different approaches utilizing FUS therapy demonstrate consistent findings, in that FUS therapy generally improves parkinsonian symptoms with only minor transient side effects. Furthermore, all of these approaches highlight the effectiveness of FUS therapy in noninvasively improving patient quality of life.

### 3.2. FUS Gene Therapy Approaches

Gene therapy is a therapeutic approach that aims to genetically modify cells through transcription and/or translation of transferred genetic material and/or integration into host genomes [80]. With PD, the goal of gene therapy is to treat disease symptoms and, ideally, to reverse disease progression. Gene therapy has been studied as a potential therapeutic for PD and other neurodegenerative diseases for years. Trophic factors, such as glial cell line derived neurotrophic factor (GDNF) and neurturin (NTN) have been explored as potential agents for PD gene therapy [81,82]. GDNF shows promise as a gene therapy agent due to its neurotrophic and neuroprotective effects. In primate models, the overexpression of neuroprotective agents such as GDNF has been demonstrated to decrease symptom severity and slow PD progression [83]. NTN is a structural and functional analogue of GDNF that has also demonstrated its ability to improve dopaminergic activity in animal models of PD [82]. Although preclinical trials provide ample evidence supporting GDNF and NTN gene therapy for the treatment of PD, clinical trials to date have not proven successful [84]. One of the key issues hindering the success of previous clinical trials has been low volumetric coverage of the gene therapy to targeted areas after direct injection [85]. In the CERE-120 clinical trial, which directly injected NTN into the putamen of study subjects, histological assessments show that the distribution of NTN was restricted [82]. Implementing FUS along with gene therapy has the potential to eliminate this issue that direct injection presents. Preclinical studies have used FUS to deliver vectors to specific animal models with encouraging results [86,87]. Xhima et al., for example, used recombinant adeno associated virus serotype 9 (AAV9) along with FUS to enhance delivery of an α-synuclein gene silencing short hairpin RNA sequence in mice overexpressing α-synuclein. FUS was targeted in the hippocampus, substantia nigra, olfactory bulb, and dorsal motor nucleus of the vagus. Decreased α-synuclein immunoreactivity was reported in these targets one month after FUS. Importantly, tyrosine hydroxylase (the rate-limiting enzyme in norepinephrine, epinephrine, and dopamine synthesis) and synaptophysin expression was not altered in targeted brain regions [86]. Mead et al. used a nanovector, particularly brain penetrating nanoparticles (BPN), along with FUS to promote GDNF transgene expression in target brain areas of rats. After only one treatment, there were therapeutically relevant levels of GDNF in targeted brain tissue. Furthermore, these therapeutic levels persisted for 10 weeks [88]. As previously mentioned, another key feature in the pathophysiology of PD is an elevated state of oxidative stress. The pro-oxidative environment established in PD potentiates neuronal stress, a pro-death environment, and progression of the disease [89]. In one study, FUS was used to deliver Nrf2, a nuclear factor which promotes downstream antioxidative elements, into the brains of PD rat models [90]. Though motor and behavioral outcomes were not measured in this study, Nrf2 expression was significantly elevated and led to a reduction in the pro-oxidative superoxide dismutase. Taken together, this suggests that FUS with viral vectors or nanovectors may show promise in developing gene therapy approaches for patients with PD. However, human evidence is lacking currently and requires more investigation.

## 4. Other Approaches in the Management of MSA and DLB

Though surgical interventions are poorly understood in the context of MSA and DLB, several promising frontiers of treatment have recently come to light. For example, in MSA, autonomic dysfunction remains a critical issue, characterized by progressive sympathetic failure, namely in the form of orthostatic hypotension [91]. This orthostatic hypotension can lead to a number of downstream sequelae, including cerebral hypoperfusion and increased risk for falls from resulting dizziness [91]. In fact, autonomic dysregulation is a predictor of worse outcomes in MSA and faster disease progression [92]. One recent study is poised to target this debilitating comorbidity of MSA [93]. This study describes the implantation of an epidural thoracic cord stimulator in a 48-year-old woman with progressive sympathetic dysfunction with resultant orthostatic hypotension. This stimulator is paired with an accelerometer which detects when the patient stands up and controls the delivery of stimulation to the thoracic cord. With the stimulator off, an 85 mmHg drop in systolic blood pressure was observed within 3 min of tilting the patient upright. This is compared to an 85 mmHg drop in systolic blood pressure occurring over 10 min after the stimulation was turned on [93]. Though this study lacks the power of a large clinical trial, it represents a promising treatment option for a very debilitating comorbidity of MSA.

In DLB, pathology stems in part from the deposition of extracellular synuclein, which leads to dysfunctional synaptic transmission and plasticity [33]. Transcranial direct current stimulation (tDCS) may play a role in modulating cortical excitability and is thought to induce changes in synaptic plasticity [94]. One recent double-blind clinical trial tested the efficacy of 10 days of tDCS sessions in improving the cognitive and psychiatric assessments of 11 DLB patients versus sham tDCS. Though there were no adverse effects from the treatment, no significant cognitive or psychiatric differences were found between the groups [87].

## 5. Conclusions

In this review, we have described the epidemiology, clinical presentation, molecular mechanisms, and standard of care for members of the synucleinopathy family of diseases: PD, MSA, and DLB. We examined the role of DBS in PD, including its potential for significant side effects, with special attention towards both how it is affected by and how it influences synuclein aggregation. Neuroinflammation, oxidative stress, and plasticity are central concepts in the pathogenesis of synucleinopathies. In PD, DBS may play a neuroprotective role, but these mechanisms are still poorly understood and require further investigation, as does the role of DBS in MSA and DLB. Furthermore, we examined the mechanism of FUS in opening the BBB and enhancing drug delivery, inducing thermoablation, and delivering gene therapy vectors. In summary, FUS represents a novel treatment paradigm and a promising area of future study. Finally, though they represent difficult pathological entities to treat, we further covered some novel treatment strategies for MSA and DLB.

## Figures and Tables

**Figure 1 biomedicines-10-02657-f001:**
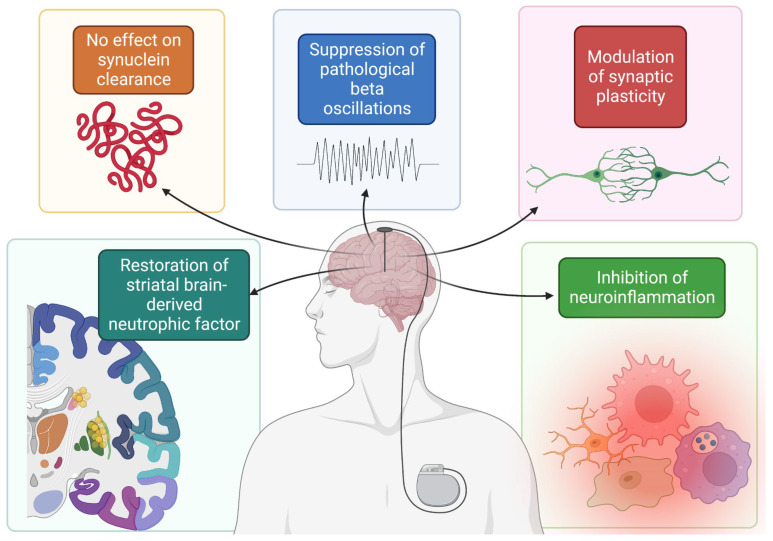
Summary of mechanisms by which DBS possibly confers a neuroprotective effect. Figure created with BioRender.com, accessed on 19 September 2022.

**Figure 2 biomedicines-10-02657-f002:**
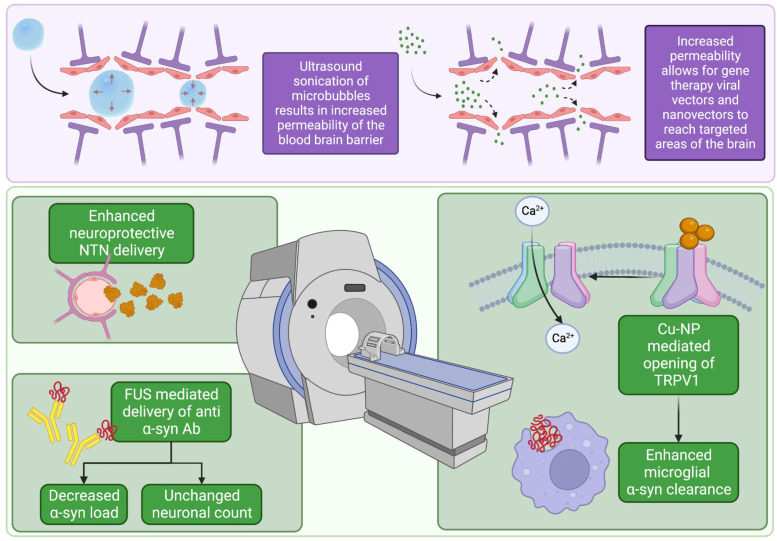
Summary of applications of focused ultrasound to synucleinopathies. Figure created with BioRender.com, accessed on 19 September 2022.

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
