# Peer review of "Surgical Management of Synucleinopathies"

_biomedicines, 2022, doi:10.3390/biomedicines10102657_

Round 1

Reviewer 1 Report

1. The review is focused on a much-needed literature survey and covers the useful aspects of treating different forms of synucleinopathies. 

However, there are many sentences with grammatical and spelling errors and making it hard to read the flow of the paper. I would strongly suggest the authors read the whole manuscript a couple of times more and fix these errors before this is accepted. Some examples are as follows:

A.  Considering Parkinson’s alone is projected to double in incidence rate by 2030 due to an aging population, it is essential to elucidated optimal management of ɑ-synucleinopathies.

B. Using animal models, accumulations of ɑ-synuclein have been identified in regions of the brain inhibiting complex I of the mitochondria

C. While more evidence as to the molecular mechanism is needed, ɑ-synuclein toxicity is well supported to play a significant role in the pathogenesis of PD

D. General risk factors for DLA include depression, anxiety, low caffeine intake, and stroke, as well a genetic predisposition with specific APOE ɛ4 alleles. What is DLA?

E. Similarly, the precise pathogenesis of DLB unknown, but is theorized to involve several metabolic pathways leading to dysfunctional mitochondria, purine metabolism, protein synthesis, energy metabolism, and ɑ-synuclein deposits in neuronal Lewy bodies. - 

F. Instead, the disease is often managed by nonpharmacologic strategies such as deceased salt intake for orthostatic hypotension

G. Namely, in this study, motor performance after 3 weeks of stimulation was significantly improved compared to rats in the nonstimulation group, even with stimulation off during motor testing. What do you mean by namely?

H. Namely, brain derived neurotrophic factor (BDNF) has come under interest recently for its roles in plasticity, neurogenesis, and neuroprotection. Similar concern?

There are more sentences other than what I have pasted here.

2. While explaining DBS, only the electrode implantation is mentioned, there is no mention or explanation of the stimulation?

Rest all looks fine.

Author Response

  1. Thank you for pointing out these insightful areas where grammar / sentence structure needed to be modified. We have combed through the manuscript extensively and have addressed all areas mentioned by the reviewer, and also made several other modifications to sentences to maintain flow.
  2. Thank you for pointing this out. Please see the new additional sentences in the “DBS and a-synuclein in PD” subsection addressing stimulation delivered through electrodes.

Reviewer 2 Report

I have no edits. This is a lovely review of a timely topic with a balanced treatment, nice coverage of the literature and excellent figures.

Author Response

Thank you for your comment! We really enjoyed working on this article.